# Sanity through Insanity: The Use of Dark Humor among United States Veterans

**DOI:** 10.3390/bs14080679

**Published:** 2024-08-05

**Authors:** Stephen M. Yoshimura, Gregory Bilbrey, Stevi A. Johns, Kristin Hall, Nathan Moore

**Affiliations:** Department of Communication Studies, University of Montana, Missoula, MT 59812, USA

**Keywords:** humor, dark humor, gallows humor, veterans, subjective well-being, life satisfaction, social relationships, social connections

## Abstract

Humor is generally known to effectively help individuals manage distress. Yet a variety of ways exist to engage in humor, and not all of them may be equally associated with desirable outcomes. The purpose of this study is to examine the extent to which dark humor is associated with the subjective well-being of United States military veterans. An online survey was announced on several social media pages populated by US veterans, to which 93 fully responded. Our findings indicate that the use of dark humor does not appear to be associated with a sense of connectedness, but the reported use of self-defeating types of dark humor was associated with lower levels of life satisfaction. Our hypothesis that increased feelings of connectedness to civilian and veteran/active-duty friends would predict increased reports of overall subjective well-being was supported. The implications of these findings for understanding the functions of dark humor are discussed.

## 1. Introduction

A long line of research demonstrates that using and being exposed to humor in the context of distressing situations reduces signs of physiological distress, increases positive emotional states, and reduces some negative psychological states such as depression and irritation [1,2]. Yet, many different ways of enacting humor exist, and questions remain about the extent to which all types of humor are equally adaptive in helping people cope with challenging situations.

Dark humor, also sometimes referred to as gallows humor, exemplifies the complicated nature of the issue. Dark humor is a type that “treats serious, frightening, or painful subject matter in a light or satirical way” [3]. Research consistently shows that individuals who experience physically and psychologically challenging situations as part of their employment (e.g., medical professionals, law enforcement officers, firefighters, and veterans) sometimes manage difficult experiences by making light of them. However, the findings appear to be mixed with regard to the relative effectiveness of this type of humor in improving indicators of reduced distress and improved psychological well-being.

This study examines the use of dark humor among current and former United States military personnel. Service members appreciate and use humor like everyone else, but often have a unique set of experiences that civilians are not easily able to identify with. For active-duty service members, these could include extended separations from civilian friends and family, the lack of autonomy in determining one’s day-to-day obligations, exposure to various environmental and personal hazards, and/or combat. Once discharged, veterans potentially manage a loss of contact with military colleagues, the loss of a military identity, and logistical challenges of relocation and career transitions in addition to residual effects of exposures they may have encountered during active duty [4]. The uniqueness of military duty, on one hand, can be a source of deep connection with others who share the same experiences. On the other hand, those unique experiences can also be isolating from civilian family members, spouses, friends, co-workers, and others who have little or no familiarity with those experiences.

With few exceptions, almost no research has focused on the use of dark humor among active-duty and veteran military service members to help cope with the unique challenges of military life. In fact, other researchers have recently emphasized that more research on the effective and appropriate use of humor as an active coping tactic within military culture is needed [5]. Given research indicating that dark humor can help individuals cope with uniquely difficult stressors [6], and that the ability to cope with stress and greater interpersonal resources are positively related to well-being among both military veterans and their spouses [4], the questions asked in the current study are a logical response to this gap in knowledge. We thus conducted this study to first examine the possibility that dark humor would be a source of connection with other veterans, but perhaps less so with civilians and family members. We then further test the notion that the use of dark humor is associated with greater degrees of perceived well-being among veterans.

### 1.1. The Concept of Dark Humor

Most definitions of dark or gallows (these two terms are often used interchangeably in research on this topic, as we also do in this manuscript) humor suggest that this type of humor has wide variation in its content and characteristics. For example, Obrdlik [7] defined it as “humor which arises in connection with a precarious or dangerous situation” (p. 709), and as humor that “originates in the process of social interaction and bears marks of the particular group in which it was created and accepted” (p. 716). As Watson [3] puts it, gallows humor is neither intentionally silly nor cruel, but is rather an honest, albeit transgressive, reflection of a difficult situation. Like all humorous messages, it is funny because the message presents an image that is simultaneously normal and a violation of a subjective moral perception of how things “ought to be” [8].

In a more specific way, dark humor can be thought of as a mildly challenging, “benign violation” of one’s sense of what is supposed to be. McGraw and Warren [9] propose three ways in which a violation can be perceived as benign and thus humorous: (1) the message simultaneously violates one norm but is acceptable within another, (2) the violated norm is considered at least somewhat acceptable to the receiver of the message, and (3) the receiver of the message feels psychologically distant from the violation. In a series of studies, McGraw and Warren [9] found support for their theory, for example, by observing that compared to scenarios in which no norm violation was committed, certain scenarios were more likely to provoke laughter when they were perceived as norm violations while also perceived as both hypothetical and potentially acceptable in some situations.

Although benign violation theory was developed to explain the nature of all humor, its focus on norm violations makes it particularly applicable to explaining why dark humor can be funny to some people and less so to others. Simply put, even a transgressive, norm-violating message can come across as funny when the message is transgressive but reasonably acceptable in a particular context, and when individuals have a degree of psychological distance from it. From this perspective, the reason why some individuals would not find dark humor funny is because they simply do not find the norm violation acceptable or are unable to feel psychologically distant from the violation.

### 1.2. The Composition of Dark Humor

Despite the definitions above, dark humor is difficult to distinguish from other humor partly because perceiving that some content is a benign rather than serious violation depends on very specific situational contexts and rules. Berger et al. [10] offer the following example:

A patient’s gynecologist tells a patient with a vulvar abnormality that she would benefit from using testosterone cream. She asks with apprehension, “Will I grow facial hair?” He retorts, “No, but you might grow a penis” (p. 827).

The authors recognize this example as a misguided attempt at humor that they refer to as “destructive physician-generated humor” because of its inappropriateness in light of the patient’s uneasiness about the issue. It is not that this message could not be considered humorous in any possible case, but rather that it would likely be considered unfunny by the patient in this particular context. Given the complexities of distinguishing dark humor from nonhumor or other types of humor, most studies on the use of dark humor rely on either the researchers’ or participants’ perceptions that a given act constitutes this type of humor. For example, Dangermond et al. [11] observed that firefighters broadly recognized and described their use of “Black humor, also sarcastic, gallows humor” (p. 41) and “highly macabre jokes” (p. 42) to cope with difficult aspects of their occupation. They offer one such example from a participant:

A resuscitation in a sushi restaurant, all you can eat. The gentleman didn’t make it. On our way back one of our colleagues said: ‘Well, that was all he could eat.’ We couldn’t stop laughing. It sure did take the tension away (p. 41).

Likewise, in an ethnographic study of rural police officers, Gaydeen and Phillips [12] observed a number of interactions that they interpreted as examples of humorous attempts at coping with difficult situations. They offer the following instance:

While responding to a residential suicide call, with little information to go on, two officers entered the home of the suspect. Upon entering the residence, one officer said to the second author, “Can you hear the music? It sounds like a funeral home”. The second officer responded, “Maybe he’s getting in the mood” (p. 51).

Obviously, distinguishing these examples as benign or serious violations of rules and norms (and thus as examples of humor or maliciousness) will depend on the context in which they occur.

To help manage some of the subjectivity involved in observing and distinguishing dark humor from other types, some studies have adapted self-report measures of humor styles to observe the extent to which dark types of humor might be used. For example, at least two studies [13,14] have used the cynicism, irony, sarcasm, and satire subscales of the Comic Style Markers scale to register dark humor on the premise that such styles reflect a tendency toward mockery and ridicule [15]. These styles are similar to the aggressive and self-defeating humor styles represented in the Humor Styles Questionnaire [16], both of which involve aggressive and disparaging types of humorous activity.

While Martin et al. [16] conceptually unite aggressive and self-defeating humor as negative or “injurious” types of humor (p. 52), they also distinguish between them along the dimension of self-enhancing vs. relationship-enhancing functions. They maintain that aggressive humor enhances the self by way of putting others down (thereby increasing a sense of superiority and control), whereas self-defeating humor enhances relationships by way of derogating one’s own sense of self in the interest of gaining approval and acceptance by others.

Most conceptual definitions of dark humor do not specify its functions, but the potential goals behind it are difficult to ignore. Papousek et al. [17] argue that dark humor is identifiable as the type that involves laughing at something or someone, in part to increase a sense of superiority or control. Such humor could be recognized as aggressive when directed toward others and as self-defeating humor when directed toward oneself. However, Martin et al. [16] add that self-defeating humor is particularly motivated by an interest in maintaining relationships with others, which resonates with the numerous studies demonstrating the role of dark humor in promoting bonding between people who share difficult experiences [15,18].

Both aggressive and self-defeating humor could potentially help service members cope with the unique demands involved in their occupation and beyond. Even while service members may not be able to remove the stressors they encounter, they could use dark humor to increase their sense of superiority or control (e.g., by using aggressive humor), or to enhance their sense of connectedness to others (e.g., by using self-defeating humor). Either way, enhancements to one’s sense of autonomy and control or relatedness have the potential to improve one’s sense of self-determination, and thus a sense of psychological well-being [19]. This explanation has not been directly invoked in past research on veterans’ use of humor, but it does explain why humor can have therapeutic functions for combat veterans and is a primary predictor of resilience among them [5,20].

Although the specific contents of dark humor are highly subjective, the theory and research reviewed above could lead one to suspect that dark humor is typically aggressive and offensive, but within a range of appropriateness for a given context. It is funny only in contexts in which the message is perceivably transgressive, but in a way that is benignly so. The mix of transgressiveness and appropriateness appears to give way to laughter about difficult situations, which is why a wide variety of studies suggest that dark humor may play a role in coping with challenges.

### 1.3. Dark Humor as a Coping Mechanism

Several essays and studies demonstrate that gallows humor promotes coping among those who occupy their time in difficult situations. For example, van Wormer and Boes [21] write that by definition, gallows humor “proposes an illogical, incongruous response to the most hopeless of situations and offers the person a triumph of sorts”, and that it “represents more of a philosophical attitude than a particular repertoire of jokes; *it is a way to maintain sanity under insane circumstances*” (p. 91, italics ours). Scholars have examined the use of humor as a coping mechanism in a number of different challenging occupations, including among medical professionals [3,10,21,22,23], law enforcement officers [12,18,24], firefighters [11,25], and to a surprisingly lesser extent, veterans [5,26]. In many cases, this body of research posits that humor can be a way to cope with seemingly hopeless situations by facilitating reappraisal of the situation, making them seem less challenging [23].

Other explanations are also possible, however. For example, some scholars also suggest that gallows humor can promote coping by building rapport and camaraderie, and generally enhancing social cohesion with others who share similar experiences [22,23]. From this perspective, dark humor originates from a shared, backstage, secretive code between members of a group or culture. The code is functional for the specific group and conveys a shared reality when used among the members. It is not meant to be heard by outsiders and is usually offensive to them when it is [3,23]. In some cases, this type of humor is referred to as specific to particular groups or settings (e.g., “medical humor”) because it is so commonly used within those contexts [21].

Yet, because the appreciation of dark humor depends on specific types of shared experiences and understandings, its use is also potentially isolating for those who use it. Several studies observe that dark humor is typically not taken well by outsiders. For example, in an essay on the use of humor between medical providers and patients, Berger et al. [10] argued that medical humor used by both patients and physicians in medical settings can be interpreted by the other as offensive and disrespectful. In some cases, the distance and exclusion of others is the exact point of using it. For instance, in a study of the use of dark humor among crime scene investigators, Vivona [18] showed that those who use dark humor typically understand the probability that outsiders will find the use of dark humor offensive, and sometimes use it as a type of shibboleth to identify those who belong among them and those who do not.

The increased group cohesion that develops around shared understandings of dark humor can explain why this type of humor can buffer individuals from distressing situations. According to the buffering model of social support [27], increased levels of support function to reduce stress by way of helping people reappraise challenging situations. By bringing people of like kind closer together, dark humor can provide a sense of connectedness to others and in turn generate one’s perception that the situation may not be as important as it seems, and that “I can make it through this” however challenging the situation might be. At least one study has demonstrated evidence for the possibility that dark humor helps buffer firefighters from occupation-related burnout and post-traumatic stress [25].

Nonetheless, a notable segment of dark humor research suggests that the association between using dark humor and effective coping might be somewhat more complicated than the buffering hypothesis makes it seem. For example, in a study of child exploitation investigators, Craun and Bourke [28] found that some types of dark humor such as self-deprecating, sexual innuendo, and humor at the expense of offenders were unrelated to work-related secondary stress, and that humor at the expense of victims and joking about human behavior and society were both related to increased secondary stress, increased alcohol use, decreased social support, and increased difficulty and frequency of working with disturbing media. A qualitative study of crime scene investigators [18] also observed that while many investigators reported using dark humor to positive effect, some types of investigations, including those involving child exploitation, did not seem to allow for any type of attempt at generating laughter. This appears to have put the investigators in a type of bind, “where no amount of joking could help them make sense of certain tragedies” [18]. Yet another study [14] found that participants were more likely to positively reappraise hypothetical challenging situations when they used more good-natured, benign humor and less dark humor. These findings suggest that while dark humor can sometimes feel like it helps people manage difficult situations, its benefits may be limited to certain contents or styles.

### 1.4. The Current Study

This study tests the general proposition that dark humor is associated with a sense of connectedness between veterans. In turn, we predict that a sense of overall increased connectedness will be associated with higher reports of well-being. Specifically, we predict that:

**H1:** 
*A positive association will exist between using dark humor and veterans’ sense of connectedness to their service-member peers.*


**H2:** 
*A positive association will exist between veterans’ overall sense of connectedness and their reported level of subjective well-being.*


**H3:** 
*A positive association will exist between veterans’ use of dark humor and their subjective well-being.*


The literature also hints at the possibility that dark humor might not always be understood and appreciated by others without military experience, and that dark humor might alienate others in those cases. To further examine this possibility, we ask a research question:

**RQ1:** 
*Does veterans’ use of different types of dark humor reduce their sense of connectedness to non-veteran others?*


## 2. Materials and Methods

### 2.1. Study Design

This study involved a cross-sectional online survey design with volunteer participants anonymously responding to prevalidated measures to assess their experience with the use of dark humor, their perceived level of connectedness to others, and their perceived current level of subjective well-being.

### 2.2. Participants

Respondents were asked to take part in the study only if they identified themselves as a veteran, defined in the call as “a person who has served in the military”, including active-duty, inactive-duty, and discharged service members.

### 2.3. Setting and Procedures

The study took place online. After receiving IRB approval to collect data, an announcement calling for potential volunteers was posted on multiple social media pages oriented toward US military veterans. Participants were recruited and data were collected over a 10-day period in April of 2022. The survey took approximately 10–15 min to complete and was anonymously submitted. No incentives were offered for participating in this study.

### 2.4. Outcome Measures

All measures used for data collection are reported below. We did not exclude any measures for the purposes of this study.

#### 2.4.1. Dark Humor

For this study, participants were asked to respond to 16 items in the Humor Styles Questionnaire [16] comprising the subscales for the use of self-defeating humor (i.e., “humor as a form of defensive denial, or the tendency to engage in humorous behavior as a means of hiding one’s underlying negative feelings…” [16] and aggressive humor (i.e., “…compulsive expressions of humor in which one finds it difficult to resist the impulse to say funny things that are likely to hurt or alienate others” [16]. Examples of items representing aggressive humor include, “When telling jokes or saying funny things, I am usually not very concerned about how other people are taking it”, and “Even if something is really funny to me, I will not laugh or joke about it if someone will be offended” (reverse-scored). Examples of self-defeating humor include, “I often go overboard in putting myself down when I am making jokes or trying to be funny”, and “Letting others laugh at me is my way of keeping my friends and family in good spirits”. All items were responded to on a Likert-type scale ranging from 1 (Totally Disagree) to 7 (Totally Agree). Reliability analysis indicated that both measures were internally consistent (self-defeating humor α = 0.85; aggressive humor α = 0.76).

#### 2.4.2. Subjective Well-Being

Subjective well-being was measured using the Cantril Self-Anchoring Scale [29]. This instrument consists of two items, wherein participants are first asked to “Please imagine a ladder with steps numbered from zero at the bottom to ten at the top. The top of the ladder represents the best possible life for you and the bottom of the ladder represents the worst possible life for you. On which step of the ladder would you say you personally feel you stand at this time?” Respondents then rated their present-day life on a scale of 0 to 10, 0 being the worst possible life for them and 10 being the best possible life for them. Per the original design of this instrument, participants were also asked in a second item to additionally estimate their life satisfaction in five years using the same scale. Because the scores of these two single-item measures were highly correlated (See Table 1), they were combined into one measure of subjective well-being.

#### 2.4.3. Connectedness to Others

Participants were asked to self-report how connected they felt with people in three groups: civilian friends, veteran friends, and family. Specifically, participants were asked to rate their perceived level of connectedness with each group on a Likert-type scale ranging from 1 (Completely Disconnected) to 7 (Completely Connected).

### 2.5. Bias

Attempts to control for bias were made by requiring anonymous survey submission and by using prevalidated measures to assess humor styles and subjective well-being. However, volunteer samples are nonprobability samples, and the results of this study may be affected by selection bias.

### 2.6. Sample Size

In light of the observations that (a) dark humor is relatively common in experience but rare in frequency of use and (b) the US military is a unique culture that is not always easily accessible for research purposes, we aimed to recruit as many participants as were willing to fully volunteer for the current study. After the initial calls were made for volunteers, secondary calls were made every 2–3 days for a 10-day period, by which time 93 participants had fully responded to the survey.

### 2.7. Analysis

Linear regression analyses were used to test the hypotheses and answer the research question. Incomplete surveys with missing data were excluded from the analyses. A sensitivity power analysis using G*Power Version 3.1.9.6 [30] indicated that the minimum detectable effect size when using a multiple linear regression model to test a one-tailed hypothesis (where *p* < 0.05) with two predictors and data from 93 participants at 80% power was *f*^2^ = 0.11 (*R*^2^ = 0.10). For simple linear regression models with one predictor, the analysis indicated that the minimum detectable effect size with 80% power was *f*^2^ = 0.09 (*R*^2^ = 0.08).

## 3. Results

### 3.1. Participants

The sample of 93 participants who fully completed the questionnaire included 55 (59%) male participants, 37 (40%) female participants, and 1 (1%) participant who reported another gender identity. The average age for participants was 39.25 years of age (SD = 10.44 years). Participants were asked to select their branch of service, resulting in 51 (54%) Marines, 28 (30%) Army, 7 (8%) Air Force, 6 (7%) Navy, and 1 (1%) Coast Guard. The average time of service was 9.96 years (SD = 7.02 years). Among the respondents were 87 (93%) enlisted personnel, 5 (5%) officers, and 1 (1%) Chief Warrant Officer, which is an enlisted service member that became a specialist in their field of occupation, then commissioned as an officer.

### 3.2. Main Results

The first hypothesis predicted a positive association between veterans’ use of dark humor and their sense of connectedness to other veterans/active-duty friends. This hypothesis was not supported in the overall model, *R* = 0.17, *F*(2, 90) = 1.17, *p* = 0.28. Neither the reported use of aggressive humor, β = 0.24, *p* = 0.20, 95% CI [−0.12, 0.60] nor self-defeating humor, β = −0.15, *p* = 0.21, 95% CI [−0.38, 0.09] were related to veterans’ sense of connectedness to other veterans/active-duty friends. To address RQ1, which inquired into the potential negative association between veterans’ use of dark humor and their sense of connectedness to non-veteran others, the same model was tested for family members and civilian friends. No significant associations emerged in the models for either family members, *R* = 0.22, *F*(2, 90) = 2.21, *p* = 0.12, or civilian friends, *R* = 0.18, *F*(2, 90) = 1.46, *p* = 0.24.

The second hypothesis predicted a positive association between veterans’ sense of connectedness to veteran/active-duty friends and their reported sense of well-being. To test this hypothesis, a simple linear regression model was created using connectedness to veteran/active-duty friends as the independent variable and overall life satisfaction as the dependent variable. The results (Table 2) indicate support for this hypothesis, *R* = 0.25, *R*^2^ = 0.06, *F*(1, 91) = 5.86, β = 0.33, *p* < 0.05, 95% CI [0.06, 0.60]. To further explore potential associations between connectedness to civilian friends and family and overall life satisfaction, two additional simple linear regression models were tested. The results indicated that connections to civilian friends were also positively associated with overall life satisfaction, *R* = 0.22, *R*^2^ = 0.05, *F*(1, 91) = 4.50, *p* < 0.05, β = 0.23, *p* < 0.05, 95% CI [0.02, 0.45], but that connectedness to family was unrelated, *R* = 0.19, *R*^2^ = 0.04, *F*(1, 91) = 3.50, *p* = 0.06, β = 0.23, *p* = 0.06, 95% CI [−0.01, 0.48].

The third hypothesis predicted that veterans’ use of dark humor would be positively associated with their sense of subjective well-being. A multiple regression model with overall life satisfaction as the dependent variable and both aggressive and self-defeating humor as the independent variables was tested. The results (Table 2) indicated overall support for the model, *R* = 0.41, *R*^2^ = 0.17, *F*(2, 90) = 8.93, *p* < 0.001. However, examination of the individual coefficients indicated that whereas the increased reported use of self-defeating humor was negatively associated with overall well-being, β = −0.60, *p* < 0.001, 95% CI [−0.89, −0.31], the reported use of aggressive humor was unrelated (β = 0.05, *p* = 0.84, 95% CI [−0.40, 0.49].

### 3.3. Supplementary Analyses

Observing that self-defeating humor and connectedness to veteran/active-duty friends and civilian friends were each significant predictors of subjective well-being, we further examined and compared the contributions of these two factors to this outcome by combining them into one multiple-regression model. The results indicated that the overall model explained 22% of the variance, *R* = 0.46, *R*^2^ = 0.22, *F*(3, 89) = 8.13, *p* < 0.001. However, the only significant predictor of life satisfaction in this model was self-defeating humor, β = −0.55, *p* < 0.01, 95% CI [−0.82, −0.27].

In sum, the results indicate that when veterans reported increased connectedness to both veteran/active-duty and civilian friends, they also reported greater degrees of subjective well-being. Yet, the reported use of dark humor was unrelated to a sense of connectedness to any group of contacts, and the increased use of self-defeating humor was related to less subjective well-being overall. This latter finding held when we examined the association of self-defeating humor on subjective well-being against the association between connectedness to veteran/active-duty friends and civilian friends and the same.

## 4. Discussion

The purpose of this study was to examine the possibility that the use of dark humor could be a means of increasing connectedness and a sense of subjective well-being among United States military service members. Past research has effectively demonstrated that individuals in a wide variety of contexts use dark humor as a means of talking about difficult situations. However, while several essays suggest that dark humor has the potential to serve as a means of effectively coping with those situations [3,23], the empirical evidence supporting that idea appears to be mixed. This study’s findings indicate some of the ways in which the connection between dark humor and subjective well-being can be complicated.

First, no support emerged for the hypothesis that the increased use of dark humor would positively relate to a sense of connectedness to other veterans/active-duty friends, civilian friends, or family members. On one hand, this finding contradicts the notion that humor can help increase a sense of group cohesiveness by demonstrating shared understandings [2]. If dark humor were encoded and decoded as a way of promoting shared identity, we should have at least found that veterans who reported using more dark humor would have reported an increased sense of connection to other service members. Importantly, however, even while dark humor was unrelated to increased connectedness between veterans and any other group of people, it also did not appear to alienate others without military experience. While dark humor could serve other important functions related to coping that we did not examine in this study (e.g., reappraisal, control, or entertainment, among others), a sense of solidarity with others apparently was not affected by it either way.

As discussed in the rationale for this study, past research suggests that even while dark humor can have a coping function and is understood as funny by people who share experiences and understandings, some boundaries still exist around the types of dark humor content that could be considered appropriate for the situation [11,18,31]. Benign violation theory would explain that messages become less funny to the extent that they are perceived as less benign, and more inappropriate violations of social rules. By extension, it stands to reason that perhaps the specific types or contents of dark humor can leave recipients with variable interpretations and impressions of the user’s motivations and thus affect how likely they are to respond in positive or negative ways.

Second, veterans who reported an increased sense of connectedness also reported higher levels of subjective well-being. This finding aligns with a vast body of research that convincingly demonstrates that both quantity and quality of social relationships are beneficial to the well-being of everyone, veterans included [32,33]. Although this finding initially might not seem to add much to current knowledge about the benefits of social connectedness, it becomes more intriguing when paired with the finding that increased use of self-defeating humor was related to lower levels of subjective well-being. Together, these two findings hint that even when connected to others, the use of self-defeating humor is involved in a feeling of reduced life satisfaction among veterans.

To further examine this issue, we compared the contributions of self-defeating humor against connectedness to veteran/active-duty friends and civilian friends to see which variables had stronger contributions to overall life satisfaction, discovering that self-defeating dark humor was the only variable among this set that remained associated with life satisfaction. We surmise that even when connected to others, service members—like most other people—can sometimes feel dissatisfied with life and use self-defeating humor as a way to manage those feelings. However, without the ability to conduct a causal analysis, it remains unclear if self-defeating humor is a cause or consequence of reduced subjective well-being. If it is a cause, then perhaps self-defeating humor suppresses other factors that would otherwise increase life satisfaction. If a consequence, then this type of humor might simply reflect one’s feelings about how well things are going. Further longitudinal research should address these issues. 

## 5. Limitations and Conclusions

Using self-report questionnaire methods announced over social media, this study gathered data from a volunteer sample of 93 US military veterans about their use of dark humor, their sense of connectedness to others, and their current level of life satisfaction. With these methodological procedures come some reasons to be cautious while interpreting the findings of these studies. First, the sensitivity power analysis suggested that the analyses may be underpowered if one considers a detectable effect size of *R*^2^ = 0.10 to be larger than a practically or theoretically relevant effect size [34]. Larger sample sizes are desirable because they increase the likelihood that a finding could be replicated [35,36]. Given that little to no known research exists on the use of dark humor and the well-being of military service members, however, the current findings may still offer descriptive and heuristic theoretical value even while also requiring further testing for replication.

Second, although the sample is composed of US veterans from almost all branches of the military and with widely variable rank and years of service, most were veterans of the US Marine Corps and Army. The fact that all of our participants were volunteers and primarily from those two branches and with no civilians to compare them against generally limits our findings to veterans from those branches who, for unknown specific reasons, not only used social media extensively enough to see the announcement for the study but also willingly agreed to participate in this study and completed the entire survey. This group of participants may not fully represent all US military service members, and we are also unable to determine the extent to which this particular sample is comparable to non-service members. Future research on veterans’ use of dark humor would do well to add other groups to compare veterans against to help determine the extent to which the use of dark humor is unique among veterans, or simply part of a trait-based humor style that may or may not have existed prior to enlisting in military service.

Third, the measures we used to gather data on the use of dark humor were originally designed to measure humor styles [16], and while those measures are able to broadly capture one’s propensity toward engaging in humor that could be considered dark humor, these self-report measures might offer only a partial look at the wide array of different contents that dark humor could potentially have. No known specific measure of dark humor currently exists, and other recent research has used similar types of measures to gather self-report data on the use of dark humor [13,14,37]. Yet, the lack of a specific measure designed to assess the contents and use of dark humor obviously limits the conclusions we can make about it.

In addition, some questions have been raised about the construct and discriminant validity of the HSQ as a whole [38,39,40], although these concerns appear to apply less to the self-defeating humor and aggressive humor measures used in this study than to the self-enhancing and affiliative humor style measures, which were not administered in the current study. The concerns around the conceptual and statistical overlap between various humor styles highlight the importance of further research designed to develop a measure for the use of dark humor specifically.

Despite these limitations, this study’s findings add to current knowledge by showing that the use of dark humor is neither a means of connection to other veterans nor a means of alienation from civilian friends and family. In addition, the findings suggest that the type of dark humor that involves self-deprecation is unlikely to be effective in helping US military service members adjust to overall life conditions, even when they feel connected to others. Nonetheless, feeling increasingly connected to other veteran/active-duty and civilian friends was related to higher levels of subjective well-being, suggesting that building good quality connections with friends is a means by which veterans thrive, whether through the use of dark humor or otherwise.

## Figures and Tables

**Table 1 behavsci-14-00679-t001:** Descriptive statistics for all measures.

Variable	2	3	4	5	6	7	8	9	10	Mean (SD)	α
1. Age	0.48 **	0.15	−0.19	0.09	−0.11	−0.05	0.19	0.02	0.11	39.25 (10.44)	
2. Years in Service	-	−0.04	−0.11	−0.01	−0.05	0.03	0.03	−0.04	−0.01	9.96 (7.02)	
3. Connectedness to civilian friends		-	0.38 **	0.49 **	−0.11	−0.16	0.19	0.21 *	0.22 *	4.35 (1.62)	
4. Connectedness veteran/active-duty friends			-	0.20	0.10	−0.10	0.14	0.32 **	0.25 **	5.75 (1.30)	
5. Connectedness to family				-	0.08	−0.17	0.15	0.21 *	0.19	5.14 (1.45)	
6. Aggressive humor					-	0.28 **	−0.13	−0.05	−0.10	4.08 (0.78)	0.76
7. Self-defeating humor						-	−0.37 **	−0.38 **	−0.41 **	4.12 (1.19)	0.85
8. Present life-satisfaction							-	0.72 **	0.92 **	6.99 (1.85)	
9. Future life-satisfaction								-	0.93 **	8.68 (1.91)	
10. Overall life-satisfaction									-	7.83 (1.74)	0.84

Note: * *p* < 0.05, ** *p* < 0.01.

**Table 2 behavsci-14-00679-t002:** Regression models predicting subjective well-being.

Independent Variable	*R*	*F*	*R* ^2^	β	SE	95% CI [LL, UL]
Connectedness to veteran/active-duty friends	0.25	5.86 *	0.06	0.33	0.13	0.06, 0.60
Connectedness to civilian friends	0.22	4.50 *	0.05	0.23	0.11	0.02, 0.45
Connectedness to family	0.19	3.50	0.04	0.23	0.12	−0.01, 0.48
	0.40	8.94 ***	0.17			
Aggressive humor				0.05	0.23	−0.40, 0.49
Self-defeating humor				−0.60 ***	0.15	−0.89, −0.31

Note: beta coefficients are unstandardized; * *p* < 0.05, *** *p* < 0.001.

## Data Availability

The data presented in this study are available upon request from the corresponding author for ethical reasons.

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
