# Peer review of "Sanity through Insanity: The Use of Dark Humor among United States Veterans"

_behavsci, 2024, doi:10.3390/bs14080679_

Round 1

Reviewer 1 Report

Comments and Suggestions for Authors

The manuscript “Sanity Through Insanity: The Use of Dark Humor Among United States Veterans” is well written and deals with a relevant topic in humour research using an understudied population, namely veterans. The methods and analyses are adequate.

I have several comments that would help to further improve the manuscript:

1.       When reading the title and introduction, I was expecting a measure specific to dark/gallows humour (which is a narrow humour style within the mockery family). However, as the two maladaptive humour styles of the HSQ were measured, I think referring to the humour concepts employed in the study as “maladaptive humour” would be more in line with the use of these terms in humour research more generally, and with Martin et al.’s terminology more specifically.

2.       Recent studies cast doubt on the construct validity of the self-defeating scale (for an overview, see Galloway, 2023, https://doi.org/10.1515/humor-2023-0086). Given the discussion of the potential detrimental effects of this humour style, considering this limitation would be important for the present study.

3.      I request that the authors add a statement to the paper confirming whether, for all experiments, they have reported all measures, conditions, data exclusions, and how they determined their sample sizes. The authors should, of course, add any additional text to ensure the statement is accurate. This is the standard reviewer disclosure request endorsed by the Center for Open Science [see http://osf.io/hadz3]. I include it in every review.

Author Response

Reviewer 1

The manuscript “Sanity Through Insanity: The Use of Dark Humor Among United States Veterans” is well written and deals with a relevant topic in humour research using an understudied population, namely veterans. The methods and analyses are adequate.

  • Thank you for taking the time to read and comment upon our manuscript. We appreciate your thoughtful suggestions.

I have several comments that would help to further improve the manuscript:

  1. When reading the title and introduction, I was expecting a measure specific to dark/gallows humour (which is a narrow humour style within the mockery family). However, as the two maladaptive humour styles of the HSQ were measured, I think referring to the humour concepts employed in the study as “maladaptive humour” would be more in line with the use of these terms in humour research more generally, and with Martin et al.’s terminology more specifically.
  • We have carefully considered this suggestion. We ultimately decided to continue referring to the concept as dark humour, however, partly because we note the editor’s point that the positive/negative dimension of the HSQ is just one of the two relevant dimensions here, and that aggressive and self-defeating humour styles are also distinguishable as either self or relationship enhancing (Martin et al., 2003). We have now further elaborated on this distinction on pp. 3-4 of the manuscript. We agree that dark humour is not only and entirely negative and maladaptive, but rather could also have relationship or self-enhancing effects, which are presumably at least somewhat positive and adaptive in function. We have tried to highlight this idea in section 1.3 of the manuscript (pp. 4-5), and note that some studies have indeed shown that self-defeating humour, in particular, can have adaptive benefits (e.g., Heintz & Ruch, 2018). In light of these observations, we believe that referring to the concept only as maladaptive would somewhat mischaracterize the type of humour we are attempting to address in this study.
  1. Recent studies cast doubt on the construct validity of the self-defeating scale (for an overview, see Galloway, 2023, https://doi.org/10.1515/humor-2023-0086). Given the discussion of the potential detrimental effects of this humour style, considering this limitation would be important for the present study.
  • We saw that Reviewer 2 also raised this point, and read further about the concerns around this measure. We have now discussed these concerns around the validity of the HSQ on p. 11 of the manuscript where we write:

“In addition, some questions have been raised about the construct and discriminant validity of the HSQ as a whole, although these concerns appear to apply less to the self-defeating humor and aggressive humor measures used in this study than to the self-enhancing and affiliative humor style measures, which were not administered in the current study (Galloway, 2023; Heintz & Ruch, 2015; Heintz & Ruch, 2016). Concerns around the conceptual and statistical overlap between various humor styles highlights the importance of further research designed to develop a specific measure for the use of dark humor.”

  1. I request that the authors add a statement to the paper confirming whether, for all experiments, they have reported all measures, conditions, data exclusions, and how they determined their sample sizes. The authors should, of course, add any additional text to ensure the statement is accurate. This is the standard reviewer disclosure request endorsed by the Center for Open Science [see http://osf.io/hadz3]. I include it in every review.
  • We support open science principles, and have now added the following statements on p. 7:

“The sample size was not determined in advance, and further data did not exist to add to the analyses….” and, “All measures used for data collection are reported below. We did not exclude any measures for the purposes of this study.

Thank you again for taking the time to contribute your review and commentary.

Reviewer 2 Report

Comments and Suggestions for Authors

The present manuscript examines individual differences in humor styles with regard to subjective well-being and feelings of connectedness to others in a sample of 93 veterans.

I found the study topic interesting and applaud authors for their aspirations they present in their study. However, my reading indicated major concerns that cannot be remedied by a revision of the manuscript. I will discuss the main points that led me to recommend rejecting the manuscript for publication in Behavioral Sciences below.

(1) The sample size is too small for correlational research because there is random fluctuation in the findings with such an N and, thus, no robust conclusions can be derived from the present data (for a discussion see Schönbrodt & Perugini, 2013). Sample sizes with N > 250 are needed to derive findings from this line of research that allows for stable parameter estimates.

Schönbrodt, F. D., & Perugini, M. (2013). At what sample size do correlations stabilize? Journal of Research in Personality, 47(5), 609-612.

(2) The second major concern regard the assessment of "humor styles." While the Humor Styles Questionnaire (HSQ) is one of the most frequently used questionnaires in the field, there is strong evidence on the lack of validity of the questionnaire (e.g., missing discriminant validity to outcomes such as well-being). Numerous studies have shown the issues with the HSQ repeatedly using multiple methods (see the works by Heintz and Ruch on this topic).

Heintz, S. (2017). Do others judge my humor style as I do? European Journal of Psychological Assessment.

Heintz, S. (2017). Putting a spotlight on daily humor behaviors: Dimensionality and relationships with personality, subjective well-being, and humor styles. Personality and Individual Differences, 104, 407-412.

Heintz, S., & Ruch, W. (2015). An examination of the convergence between the conceptualization and the measurement of humor styles: A study of the construct validity of the Humor Styles Questionnaire. Humor, 28(4), 611-633.

Heintz, S., & Ruch, W. (2018). Can self-defeating humor make you happy? Cognitive interviews reveal the adaptive side of the self-defeating humor style . Humor, 31(3), 451-472.

Ruch, W., & Heintz, S. (2016). The German version of the Humor Styles Questionnaire: Psychometric properties and overlap with other styles of humor. Europe's journal of psychology, 12(3), 434.

Ruch, W., & Heintz, S. (2017). Experimentally manipulating items informs on the (limited) construct and criterion validity of the Humor Styles Questionnaire. Frontiers in Psychology, 8, 244621.

Taking these major points of the sample size and issues regarding the assessment of the study variables into account, I recommend rejection of the manuscript.

Author Response

Reviewer 2

The present manuscript examines individual differences in humor styles with regard to subjective well-being and feelings of connectedness to others in a sample of 93 veterans.

I found the study topic interesting and applaud authors for their aspirations they present in their study. However, my reading indicated major concerns that cannot be remedied by a revision of the manuscript. I will discuss the main points that led me to recommend rejecting the manuscript for publication in Behavioral Sciences below.

  • Thank you for your careful review of our work. The critiques, insights, and suggested citations helped us gain further insight into the weaknesses of this particular study and have learned about several issues that will help us more effectively advance this research program.

(1) The sample size is too small for correlational research because there is random fluctuation in the findings with such an N and, thus, no robust conclusions can be derived from the present data (for a discussion see Schönbrodt & Perugini, 2013). Sample sizes with N > 250 are needed to derive findings from this line of research that allows for stable parameter estimates.

Schönbrodt, F. D., & Perugini, M. (2013). At what sample size do correlations stabilize? Journal of Research in Personality, 47(5), 609-612.

  • We acknowledge the potential problems with a small sample size and have addressed this issue in two places in the manuscript. In the Method section (pp. 6-7), we now report the results of a sensitivity power analysis, refer to the reference above, and discuss how the final sample size was arrived at. We now write:

“A sensitivity power analysis using G*Power Version 3.1.9.6 (Faul et al., 1997) indicated that the minimum detectable effect size when using a multiple linear regression model to test a one-tailed hypothesis (where p < .05) with two predictors and data from 93 participants at 80% power was f2 = .11 (R2 = .10). For simple linear regression models with one predictor, the analysis indicated that the minimum detectable effect size with 80% power was f2 = .09 (R2 = .08). Although larger sample sizes are desirable because they increase the likelihood that a finding could be replicated (Lakens & Evers, 2014; Schönbrodt & Perugini, 2013), we relied on volunteer sampling procedures given a set of logistical, ethical, temporal, and financial constraints, and were unable to increase the sample size in light of the boundaries of those resources. The sample size was not determined in advance, and further data did not exist to add to the analyses.”

  • We further address the issue in the Limitations and Conclusions section (p. 10) and highlight the potential for the findings to hold heuristic theoretical value despite the observation that the sample size falls below what Schönbrodt and Perugini found to be the point of stability in the context in which they compared their correlations. Although the sensitivity power analysis suggests that the analyses are underpowered, the lack of previous research in this specific area leaves open questions about what the true and the smallest relevant effect sizes of the association between dark humor, connectedness, and well-being could potentially be. We thus reason that the findings may still hold value in their potential to prompt further research.

Specifically, we now write on p. 10:

… the small sample size is reason to be cautious when interpreting the current findings, as the sensitivity power analysis suggested that the analyses may be underpowered if one considers a detectable effect size of R2 = .10 to be larger than a practically or theoretically relevant effect size (Lakens, 2022). Given that little to no known research exists on the use of dark humor and well-being of military service members, however, the current findings may still offer descriptive and heuristic theoretical value even while also requiring further testing for replication. “

(2) The second major concern regard the assessment of "humor styles." While the Humor Styles Questionnaire (HSQ) is one of the most frequently used questionnaires in the field, there is strong evidence on the lack of validity of the questionnaire (e.g., missing discriminant validity to outcomes such as well-being). Numerous studies have shown the issues with the HSQ repeatedly using multiple methods (see the works by Heintz and Ruch on this topic).

Heintz, S. (2017). Do others judge my humor style as I do? European Journal of Psychological Assessment.

Heintz, S. (2017). Putting a spotlight on daily humor behaviors: Dimensionality and relationships with personality, subjective well-being, and humor styles. Personality and Individual Differences, 104, 407-412.

Heintz, S., & Ruch, W. (2015). An examination of the convergence between the conceptualization and the measurement of humor styles: A study of the construct validity of the Humor Styles Questionnaire. Humor, 28(4), 611-633.

Heintz, S., & Ruch, W. (2018). Can self-defeating humor make you happy? Cognitive interviews reveal the adaptive side of the self-defeating humor style . Humor, 31(3), 451-472.

Ruch, W., & Heintz, S. (2016). The German version of the Humor Styles Questionnaire: Psychometric properties and overlap with other styles of humor. Europe's journal of psychology, 12(3), 434.

Ruch, W., & Heintz, S. (2017). Experimentally manipulating items informs on the (limited) construct and criterion validity of the Humor Styles Questionnaire. Frontiers in Psychology, 8, 244621.

  • Thank you for pointing out the questions that exist around the validity of this measure and for referring us to these studies. These references and others we learned about in our review of them reveal a compelling debate around the HSQ. However, we note that Heintz and Ruch (2015), in particular, found evidence of good discriminant and convergent validity for the aggressive and self-defeating humor style sub-measures (discussed on pp. 620-621), which were the only two sub-measures of the HSQ used in this study. Thus, we now discuss the matter in the Limitations and Conclusions section (p. 10), where we write:

“In addition, some questions have been raised about the construct and discriminant validity of the HSQ as a whole, although these concerns appear to apply less to the self-defeating humor and aggressive humor measures used in this study than to the self-enhancing and affiliative humor style measures, which were not administered in the current study (Galloway, 2023; Heintz & Ruch, 2015; Heintz & Ruch, 2016). The concerns around the conceptual and statistical overlap between various humor styles highlight the importance of further research designed to develop a specific measure for the use of dark humor.”

Thank you again for taking the time to carefully review our work and offer this insightful assessment.

Round 2

Reviewer 2 Report

Comments and Suggestions for Authors

I thank the authors for addressing my comments and revising the manuscript. However, the issue and limitations of the small sample size are not alleviated by providing a power analysis and noting this limitation explicitly. I think the replication crisis in psychology should have provided enough evidence showing the issues of small samples that provide underpowered and unstable findings that as a consequence of the former do not replicate and are no service to the field. Unless authors provide a replication in an independent sample the findings cannot be trusted*. 

Personal note: I have been there myself, got a manuscript rejected for this issue provided replication samples and initial effects did not replicate as they were a product of random fluctuation from too small sizes. While I understand the frustration of getting an ms rejected after investing many work and efforts, truth is that statistics cannot be out-played by power analyses and noting potential issues that come with small samples. 

I wish the authors the best with their project and would suggest to collect more data to save this project. 

Author Response

Thank you again for taking the time to review the revised version of our manuscript. Below are our responses to the commentary.

Original comment: I thank the authors for addressing my comments and revising the manuscript. However, the issue and limitations of the small sample size are not alleviated by providing a power analysis and noting this limitation explicitly. I think the replication crisis in psychology should have provided enough evidence showing the issues of small samples that provide underpowered and unstable findings that as a consequence of the former do not replicate and are no service to the field. Unless authors provide a replication in an independent sample the findings cannot be trusted*. 

Response: We understand and are sensitive to the issues around replication and truth in research findings, and fully respect your decision on this manuscript. We have a few observations about the larger issue of the replication crisis as it applies to the current study, although we also understand that these observations may not result in a different recommendation.

(1) To the point that explicitly noting a limitation does not resolve the limitation: We agree that this limitation is real, and do not pretend that simply noting the limitation makes it go away. However, we also believe that transparency is important when the limitations are known to exist. Based on the initial set of reviews, we now make the limitation more prominent in the manuscript to allow readers make an informed decision about the findings in light of whatever knowledge they may need from them. Our goal was not to infer or recommend specific practical actions or applications from these data, nor have we attempted to make far-reaching conclusions about them. Rather, we are presenting the findings to hopefully contribute some findings to a currently understudied area on a difficult-to-reach population, even if those findings are uncertain at this time.

(2) To the point that these data are of no use: At this time, few studies yet exist on the use of dark humor among US military veterans and the extent to which this type of humor relates to well-being in this specific population. As such, we believe that there is value to published data in this area even if those data are tentative and mainly useful for descriptive purposes.

(3) Regarding the overall issue of stability in research findings: We have read the Schönbrodt and Perugini (2013) report cited in the initial review and have taken note of their findings. This is a useful resource and we are grateful to have it brought to our attention. At the same time, we believe that research findings and effect sizes should be interpreted in a larger context. The true effect sizes for the associations between dark humor, feelings of connection, and the subjective well-being of veterans are difficult if not impossible to know. No meta analyses exist to work with on this topic, let alone any meta-meta analyses as Schönbrodt and Perugini were able to apply to their question. Indeed, if such analyses already existed there would be little reason to conduct research on the matter in the first place. As such, we wonder what the smallest effect sizes of interest would be. The answer to this question is, of course, subjective. In relative terms, most effect sizes in social scientific research are likely to be "small" as defined by Cohen (1988, see also Lakens, 2022), so perhaps our statistical results are not entirely irrelevant. It is true that the possibility of Type II error is a concern with a small sample size, but that concern would still exist even if the number of participants reached the number at which Schönbrodt and Perugini found their correlations surpass what they arbitrarily determined was the "Point of Stability" (N = 161, see Figure 1) for their specific data. Even then, these authors note that determining a meaningful correlation is a subjective matter that depends on the context of a given study. In the current case, we believe that the findings at least offer descriptive and heuristic value and could prompt further interest and investigation into the issue addressed in this study. This, in our opinion, is one of the major purposes of research in the first place.

Original Comment: Personal note: I have been there myself, got a manuscript rejected for this issue provided replication samples and initial effects did not replicate as they were a product of random fluctuation from too small sizes. While I understand the frustration of getting an ms rejected after investing many work and efforts, truth is that statistics cannot be out-played by power analyses and noting potential issues that come with small samples. I wish the authors the best with their project and would suggest to collect more data to save this project. 

Response: We did not have any expectations of publication when we decided to submit this manuscript for review, and we respect and appreciate your honest assessment. We know that it takes a lot of time to provide a good review and are grateful for this. The sources you referred to in the initial review were very helpful and we will keep these in our database for future work that we hope to conduct in time.

While we understand the potential value of collecting more data, we stand by our perception that the current findings have value in their ability to prompt further questions and describe the links between this type of humor use, connectedness, and well-being among US veterans. The editor has agreed and recommended acceptance of this paper at this time based on the other anonymous review and our revisions in response to the initial reviews. However, we have taken your commentary seriously and intend to use a priori power analysis in the future to help ensure that our future findings are sufficiently statistically powered.

Thank you again for your thoughtful and helpful assessment of this manuscript.

References

Cohen, J. (1988). Statistical power analysis for the behavioral sciences (2nd ed.). Lawrence Erlbaum Associates.

Lakens, D. (2022). Sample Size Justification. Collabra: Psychology, 8(1), 33267. https:doi.org/10.1525/collabra.33267

Round 3

Reviewer 2 Report

Comments and Suggestions for Authors

Dear authors, 

thank you for carefully responding to my comments. I repeat myself, but these points do not resolve the main issue with the present manuscript.

While I enjoy the discussion about meta-science, I am shocked that we still discuss in 2024 in a third round of reviews whether an underpowered study that produces unstable effects using a strongly criticized study instrument should be published. 

Some notes on the authors' responses: 

(1) "To the point that these data are of no use" I have not noted that the data are of no use. The data are of use, especially when replicating them with additional data and, for example, using a mini meta-analysis to aggregate the findings and evaluating their robustness and checking whether they are trustworthy. What I noted, and this remains the case from a statistical point of view and probability theory, the findings of the present sample cannot be trusted. 

(2) The assumption that the present data would be a heuristic for future research is one of the main problems of the replication crisis: Using underpowered and low n studies as a benchmark or orientation for findings. Moreover, the reality is that once such findings are published they are perceived as "true" because they have been peer-reviewed. 

(3) "At the same time, we believe that research findings and effect sizes should be interpreted in a larger context. The true effect sizes for the associations between dark humor, feelings of connection, and the subjective well-being of veterans are difficult if not impossible to know. No meta analyses exist to work with on this topic," I agree with the authors and this highlights the need for a study that allows to derive somewhat stable and trustful conclusions instead of interpreting parameters that are strongly affected by random fluctuation. 

(4) "n relative terms, most effect sizes in social scientific research are likely to be "small" as defined by Cohen (1988, see also Lakens, 2022), so perhaps our statistical results are not entirely irrelevant. It is true that the possibility of Type II error is a concern with a small sample size, but that concern would still exist even if the number of participants reached the number at which Schönbrodt and Perugini found their correlations surpass what they arbitrarily determined was the "Point of Stability" (N = 161, see Figure 1) for their specific data." Please note that the question of statistical power is not the same as the stability of effect parameters. For example, a study might have enough power to detect a comparatively large effect. At the same time the n might be so low that the finding which is correctly detected will not replicate because the effect itself is a random finding because of small sample size and instability of parameter estimation. Both issues exist in the present manuscript. 

Further, I commend the notion that the findings are likely to be characterized by small effect sizes. This has been corroborated by works Fraley and Vazire (2014, 2022), showing that the majority of effect sizes in the field of psychology are small. Since we know this and accept this assumption, we should implement this knowledge in the decision on whether a study design allows to draw conclusions about the study subject or whether findings are based on underpowered and unstable parameter estimates that will likely not replicate and cannot be trusted because they are based on random fluctuation. 

Fraley, R. C., & Vazire, S. (2014). The N-pact factor: Evaluating the quality of empirical journals with respect to sample size and statistical power. PloS one9(10), e109019.

Fraley, R. C., Chong, J. Y., Baacke, K. A., Greco, A. J., Guan, H., & Vazire, S. (2022). Journal N-pact factors from 2011 to 2019: evaluating the quality of social/personality journals with respect to sample size and statistical power. Advances in Methods and Practices in Psychological Science5(4), 25152459221120217.

Author Response

We respect the reviewer’s assessment of this manuscript and share the opinion that research with larger samples is essential for many of the same reasons. Yet, we also continue to believe that research with smaller sample sizes can offer information and be worth publishing in the interest of generating new ideas and stimulating growth in awareness of certain issues among specific groups of people. Research is, by its very nature, always incomplete and even studies with large samples can have limitations that that interfere with replicability.

This response is not an attempt to change the reviewer’s assessment or be provocative, but rather an acceptance of an editorial invitation to respond and further explain our position. Thus, we respectfully ask that the current manuscript be considered in light of some additional but related issues that several researchers suggest are important when evaluating research based on issues related to sample sizes. These include the resource constraints and feasibility of collecting data, and the relative rarity of the phenomenon being studied (Anderson & Glebova, 2022; Etz & Arroyo, 2015; Lakens, 2022).

First, this study was conducted without funding of any sort and the data were completely volunteered without compensation offered to the participants. We believe that this is a potential strength of the current study. Yet it is reasonable to expect that only a limited number of participants, particularly those who belong to unique cultures like that of the US military, will volunteer to participate in research only because they want to. The lack of financial incentive to participate may have mitigated some problems in the data (see Peer et al., 2022 for an elaboration of the many concerns about data from paid online research participants), but it also likely limited the number of people willing to engage the process of research.

Time is also a genuinely scarce resource for all researchers, but for some more than others. Whereas well-funded researchers are able to buy time and assistance in collecting large samples, others are strictly beholden to economic, temporal, and institutional constraints that they cannot always control. In the current situation, adding more data is not logistically possible in the near future, even though we agree that additional data would be desirable. We have openly emphasized the cautions that should be taken in interpreting our current findings, and agree with the reviewer’s perspective that openly noting them does not make those cautions go away. But outright rejecting studies with smaller sample sizes from publication also risks limiting empirical discourse to an elite group of researchers who are well-resourced enough to gather a volume of data arbitrarily deemed necessary to participate, and potentially contributes to a rich-get-richer type of situation in any given field. We believe that this type of inequity would be as, if not even more harmful to the body of knowledge than allowing some tentative research findings in a relatively understudied area to be added to the discussion.

Second, we ask that the current study be considered in light of the relative rarity with which dark humor is used in social life. Even if it is common in experience, dark humor is probably rarely used in frequency. Thus, large volumes of data on it may be difficult to find. Most previous research on dark humor has involved qualitative methodology partly for this reason – it is relatively infrequently used and limited to specific groups of people who share unique occupational experiences and challenges. We offered a quantitative approach here in an attempt to build upon past qualitative research and test associations between theoretically and practically relevant variables that such studies have suggested are important. But it might also be that only a limited number of US military veterans will be interested and willing to participate in an online survey study on dark humor because they either do not feel an appreciation for it, or have little experience with it that they also want to share with university researchers. 

In short, sample size and statistical power of a given set of analyses is one set of criteria by which studies should be assessed, but it is not the only one. We believe that the current study has other merits by which it could be assessed and deemed appropriate for publication in the public domain. To be sure, however, the reviewer also has additional points from the latest round of reviews that we also wish to further respond to.

  • “While I enjoy the discussion about meta-science, I am shocked that we still discuss in 2024 in a third round of reviews whether an underpowered study that produces unstable effects using a strongly criticized study instrument should be published.”

-We do not intend to shock or provoke the reviewer. Our responses are required as part of the process of open peer review at this journal. In the spirit of academic debate, the studies the reviewer referred to in the initial review criticizing the Humor Styles Questionnaire (Martin et al., 2003) seem to generally point to concerns about the validity of the subscales used to assess affiliative and self-enhancing humor styles (see Galloway, 2023 for a review). These submeasures were not even used in the current study. The submeasures we did use (those for aggressive and self-defeating humor) appear to hold up respectably well to various tests of reliability and validity. Martin (2015) has also published a response to these critiques defending the scale overall. Finally, the notion that a study using a critiqued measure should not be published assumes that the critiques are entirely correct and that the entire measure has been fully invalidated. The measure has not been fully invalidated, and in fact shows convincing indications of reliability, reasonably good structure, and respectable indications some types of validity, even if some critiques are also fair to make about it.

  • "I have not noted that the data are of no use.”

-To clarify, this was our interpretation of the statement made in the first paragraph of the second review that reads, “…the replication crisis in psychology should have provided enough evidence showing the issues of small samples that provide underpowered and unstable findings that as a consequence of the former do not replicate and are no service to the field.”

  • “The assumption that the present data would be a heuristic for future research is one of the main problems of the replication crisis: Using underpowered and low n studies as a benchmark or orientation for findings.

-Perhaps we should have chosen a different word, but by “heuristic,” we mean promoting motivation for further inquiry and discovery. We believe that research findings can and should raise more questions than they answer, and thus motivate new investigations. This prompting would be useful in areas where little research has yet been conducted, such as the current one.

  • “Moreover, the reality is that once such findings are published they are perceived as "true" because they have been peer-reviewed.”

-The claim that findings will be considered as true just because they have been published in a peer-reviewed journal is speculative, and likely depends on the background, training, attentiveness, and motivations of the audience in question. That said, it is true that popular press outlets will sometimes oversimply peer-reviewed research findings and present them without the appropriate nuances. This is a separate concern, however, and not entirely within our control beyond making the nuances and limitations as clear as we could. We believe the revised version of the manuscript makes it clear that the findings need to be considered with an eye toward the statistical power of the analyses and other limitations.

Although we have some points of disagreement, we believe that open academic conversation and debate is a healthy and essential component of the peer review process. Whatever the outcome, we know that careful, involved reviewers are difficult to find, and very much appreciate both the willingness to participate in this conversation and the concern about these important issues.

References

Anderson, S. R., & Glebova, T. (2022). Bigger isn't always better: The benefits of small-sample research designs. Journal of Marital and Family Therapy, 48, 957-960. https://doi.org/10.1111/jmft.12599

Etz, K. E., & Arroyo, J. A. (2015). Small sample research: Considerations beyond statistical power. Prevention Science, 16, 1033-1036. https://doi.org/10.1007/s11121-015-0585-4

Galloway, G. (2023). The Humor Styles Questionnaire: a critique of scale construct validity and recommendations regarding individual differences in style profiles. Humor, 36(4), 631-649. https://doi.org/10.1515/humor-2023-0086

Lakens, D. (2022). Sample Size Justification. Collabra: Psychology, 8(1), 33267. https://doi.org/10.1525/collabra.33267

Martin, R. A., Puhlik-Doris, P., Larsen, G., Gray, J., & Weir, K. (2003). Individual differences in uses of humor and their relation to psychological well-being: Development of the Humor Styles Questionnaire. Journal of Research in Personality, 37(1), 48-75. https://doi.org/10.1016/S0092-6566(02)00534-2

Martin, R. A. (2015). On the challenges of measuring humor styles: Response to Heintz and Ruch. Humor, 28(4), 635-639. https://doi.org/10.1515/humor-2015-0096

Peer, E., Rothschild, D., Gordon, A., Evernden, Z., & Damer, E. (2022). Data quality of platforms and panels for online behavioral research. Behavior Research Methods, 54(4), 1643-1662. https://doi.org/10.3758/s13428-021-01694-3